# The Superior Laryngeal Nerve and Its Vulnerability in Surgeries of the Neck

**DOI:** 10.3390/diagnostics11071243

**Published:** 2021-07-12

**Authors:** Antonio S. Dekhou, Robert J. Morrison, Jickssa M. Gemechu

**Affiliations:** 1Department of Foundational Medical Studies, Oakland University William Beaumont School of Medicine, Rochester, MI 48309, USA; antoniodekhou@oakland.edu; 2Department of Otolaryngology-Head & Neck Surgery, Michigan Medicine, University of Michigan, Ann Arbor, MI 48109, USA; morrisor@med.umich.edu

**Keywords:** superior laryngeal nerve, thyroidectomy, neuroanatomy, superior thyroid artery, anatomical variations

## Abstract

Anatomical considerations of the superior laryngeal nerve (SLN), a branch of the vagus, provides information to minimize the potential for iatrogenic intraoperative injury, thereby preventing motor and sensory dysfunctions of the larynx. The present study aims to assess the variation of the SLN and its relationship to the superior thyroid artery (STA) and superior laryngeal artery (SLA). The study was done on 35 formalin-fixed cadavers at Oakland University in 2018–2019. In our study, we found that out of 21 cadavers, 52.4% of the external laryngeal branches (ebSLN) are related posteromedial to the STA, while 47.6% are related anteromedial to it. Out of 14 cadavers, 64.3% of the internal laryngeal branches (ibSLN) are related superoposterior to the SLA, while 35.7% are inferoposterior to it. In most cases, the SLA crosses above the ebSLN while traveling to pierce the thyrohyoid membrane to reach the larynx. The data demonstrate that both the ebSLN and ibSLN display variation in their relationship with the STA and the SLA, respectively. Awareness of these variable relationships is critical for identification and isolation of these structures in order to prevent consequences of nerve injury, primarily a reduction in the highest attainable frequency of the voice and aspiration pneumonia.

## 1. Introduction

The superior laryngeal nerve (SLN), a branch of the vagus nerve, arises from the inferior vagal ganglion and descends posteromedial to the internal carotid artery (ICA) before branching distally into two nerves: The smaller external branch of the superior laryngeal nerve (ebSLN) and the larger internal branch of the superior laryngeal nerve (ibSLN) [1]. Each branch of the SLN possesses a unique relationship with a respective artery: The ebSLN with the superior thyroid artery (STA) and the ibSLN with the superior laryngeal artery (SLA) [1,2]. The ebSLN, along with the STA, descends from the carotid sheath towards the superior pole of the thyroid [3]. While the STA provides blood supply to the superior pole of the thyroid gland, the ebSLN is responsible for motor innervation of the cricothyroid muscle, which acts to elongate and thin the true vocal folds during phonation, thereby elevating the pitch of the voice [1]. The larger ibSLN courses with the SLA, which is a branch of the STA. Together, the ibSLN and the SLA pierce the thyrohyoid membrane, a membrane bounded superiorly by the hyoid bone and inferiorly by the thyroid cartilage, to enter the laryngeal vestibule for sensory innervation and blood supply to the laryngeal mucosa, respectively. The ibSLN commonly trifurcates in the laryngeal vestibule, providing sensory innervation to the epiglottis, anterior wall of the vallecula, aryepiglottic folds, and motor innervation to the interarytenoid muscle [2,4].

During surgical procedures of the central and lateral neck, the SLN is at risk for injury. Damage to branches of the SLN can occur by transection, traction, thermal injury, compression, or ligation of nearby vessels during surgical procedures of the central and lateral neck [5]. Although damage to the SLN may lead to significant disability, it is commonly regarded as inconsequential due to the emphasis placed on the recurrent laryngeal nerve (RLN), which provides primary motor innervation to the vocal cords [6].

While there are techniques in surgical practice to best identify the RLN during central neck surgery, there is no clear consensus or standard technique for the identification of the SLN [7]. There is also less emphasis in the literature regarding the anatomical variations of the SLN, although it is important to recognize that significant anatomical variations of the SLN among the general population lead to increased vulnerability during surgical procedures of the neck [8]. This cadaver-based project aims to qualitatively and quantitatively assess the anatomical considerations of the SLN pertaining to its course and relationship to the STA and SLA. This information will be helpful to operating surgeons in the identification and isolation of the nerve, thereby minimizing iatrogenic injury and post-operative complications.

## 2. Materials and Methods

The study was performed on a total of 35 formalin-fixed cadavers at Oakland University William Beaumont School of Medicine in 2018–2019. Anterior cervical deep dissection was performed on each cadaver with simultaneous identification and isolation of various structures, including the common carotid artery (CCA), external carotid artery (ECA), internal carotid artery (ICA), ibSLN, ebSLN, STA, SLA, thyroid gland, thyrohyoid membrane, and cricothyroid muscle, whenever possible. Following critical observation and careful dissection, photographs were taken, and the data were analyzed quantitatively and in a descriptive manner. Specifically, we focused on the anatomical relationship of the ebSLN to the STA and the ibSLN to the SLA. Because not all cadaveric neck dissections were amenable to study both nerves, different sample sizes of the ebSLN and ibSLN were recorded. Anatomical variations of the ebSLN and ibSLN in their course and relationship with their respective arteries were also documented.

## 3. Results

Of the 35 total cadavers, 20 were male and 15 were female. Out of 21 cadavers in whom the ebSLN was dissected, 52.4% of the ebSLN are related posteromedial to the STA, while 47.6% are related anteromedial to it (Figure 1). When stratified by sex, the ebSLN is more commonly related posteromedial to the STA in male cadavers (63.6%) compared to female cadavers (40.0%) (Figure 1). Conversely, the ebSLN is more commonly related anteromedial to the STA in female cadavers (60.0%) compared to male cadavers (36.4%) (Figure 1). Out of 14 cadavers in whom the ibSLN was dissected, 64.3% of the ibSLN are related superoposterior to the SLA, while 35.7% are inferoposterior to it (Figure 2). When stratified by sex, the ibSLN is more commonly related superoposterior to the SLA in female cadavers (83.3%) compared to male cadavers (50.0%) (Figure 2). Conversely, the ibSLN is more commonly related inferoposterior to the SLA in male cadavers (50.0%) compared to female cadavers (16.7%) (Figure 2).

Several unique anatomical relationships of the ebSLN and ibSLN with the STA and SLA, respectively, are identified. Figure 3 displays the ebSLN coursing anteromedial to the STA but then inserts immediately posterior to it and simultaneously bifurcates to innervate the cricothyroid muscle. The ibSLN is shown coursing superoposterior relative to the SLA (Figure 3). In another cadaver, the ebSLN courses posteromedial to the STA (Figure 4). The ibSLN is not depicted (Figure 4). Another variation is depicted in which the ebSLN courses anteromedial relative to the STA (Figure 5). The ibSLN is shown traveling inferoposterior relative to the SLA, which is directly branching off the ECA (Figure 5). In addition, the SLA is branching directly from the ECA rather than the STA (Figure 5). There were no significant differences between right and left sided ebSLN or ibSLN on each cadaver. In most cases, the SLA crosses above the ebSLN while traveling to pierce the thyrohyoid membrane to reach the larynx.

## 4. Discussion

Both anatomical variations of the SLN and complications following SLN paralysis have not been well-studied in the extant literature relative to its counterpart, the RLN. However, the SLN has many clinical implications ranging from loss of ability to elevate voice pitch due to loss of cricothyroid motor innervation to aspiration due to loss of laryngeal cough reflex. Awareness of the anatomical variations of both the ebSLN and ibSLN with the STA and SLA, respectively, is clinically significant to minimize nerve injury during surgical procedures of the neck. A brief literature review of the variation of the SLN and its implications in surgical procedures of the neck is summarized in Appendix A [1,9,10,11,12,13,14,15,16].

The ebSLN is most commonly damaged in operations of the thyroid gland, specifically thyroidectomy [5]. Although SLN paralysis is clinically underreported and widely variable in the literature, the rate of injury is reported to be as high as 15% during thyroidectomies [17,18]. The ebSLN gained surgical notoriety when the famous Italian soprano, Amelita Galli-Curci, underwent a thyroidectomy for a large thyroid goiter, after which she suffered the loss of ability in the upper register of her singing voice, ability to sustain notes, and evident breathlessness [19]. The post-operative complication was likely due to the loss of ebSLN function caused during the surgery, which eventually prevented the star from continuing her career as a talented soprano. Additionally, damage to the ebSLN may result in hoarseness of the voice and monotonous voice production, which may be detrimental to individuals, such as singers and artists [19].

In efforts to preserve the ebSLN, the literature describes various modes of classification of the SLN, specifically of the ebSLN, in relation to its surrounding anatomic structures. The Cernea classification describes the distance, in centimeters, from which the ebSLN crosses over the superior thyroid vessels in relation to the upper lobe of the thyroid gland [20]. Other classification systems include Friedman’s and Selvan’s classification, which describe the relationship of the ebSLN to the inferior pharyngeal constrictor and the cricoid cartilage, respectively [20]. To our knowledge, however, there is no study that describes the relationship of the ebSLN and STA in directional anatomic terms along their course.

Given the close proximity of the ebSLN to the STA, it is crucial that the surgeon be cautious during ligation of the STA during central neck procedures, such as thyroidectomy, especially since it is not routinely identified and isolated as is the RLN. In a study by Ortega and colleagues, the ebSLN was anatomically described using 157 cadavers in relation to various anatomical landmarks [21]. In this study, they found that 89.6% of the ebSLN coursed medial to the STA. Similarly, in our study, all ebSLN were medial to the STA, however, the investigation also revealed that the ebSLN more commonly coursed posteromedial to the STA rather than anteromedial. However, it is crucial to emphasize the variation present upon stratification by sex, which displayed that the ebSLN was more commonly related posteromedial in male cadavers and anteromedial in female cadavers. The data from our study emphasize anatomical relationships, which helps provide information regarding the orientation of the nerve to the superior neurovascular pedicle of the thyroid gland.

The variation in the anatomical relationship of the ibSLN with the SLA, and other surrounding anatomical structures is clinically significant. Unlike the ebSLN, classification systems of the anatomic course of the ibSLN have not been previously described. The anatomical course and variation of the ibSLN are important in both the surgical setting and the clinical setting. Surgically speaking, the ibSLN is at risk for injury during procedures involving anterior approaches to the cervical spine. More specifically, the ibSLN is particularly vulnerable to stretching, compression, or ligation at the C4-C6 level, near the thyrohyoid membrane [2].

Procedures in which the ibSLN is at risk for injury include anterior cervical discectomy with fusion (ACDF), cervical lymphadenectomy, thyroidectomy, parathyroidectomy, and carotid endarterectomy [4,22]. Given that the ibSLN afferents are also responsible for the glottic closure reflex during swallowing, vomiting, and coughing, damage to this nerve increases the risk of aspiration in addition to problems with phonation due to interarytenoid muscle paralysis [4].

In a clinical setting, the anatomical variation of the ibSLN is also significant. Cough is one of the most common complaints for which patients seek medical care [23]. While there are a variety of causes and forms of cough, the ibSLN is implicated in the development of neurogenic cough, as it is responsible for the laryngeal cough reflex [2,19,20]. Neurogenic cough is thought to be due to sensory neuropathy and laryngeal hypersensitivity [24]. An emerging treatment for refractory neurogenic cough is in-office SLN blockade via injection of local anesthetic and corticosteroids [24]. The in-office SLN blockade involves palpation of anatomical landmarks to appropriately block the ibSLN along its course. More specifically, the superior thyroid tubercle and greater horn of the hyoid bone are palpated as anatomical landmarks to identify the thyrohyoid membrane, which is the entry point of the ibSLN into the larynx [24]. The anesthetic and steroid are then successfully injected towards the posterior thyrohyoid membrane, a common trigger point in patients with neurogenic cough, thereby blocking the ibSLN along its course [24]. The presence of anatomical variation of the ibSLN is particularly relevant to practitioners performing this treatment, such as otolaryngologists. As described by Kiray et al. in 2006 [2], the most vulnerable location of the ibSLN is near the thyrohyoid membrane. There have been case reports documenting adverse effects of SLN blocks, which include hypotension, bradycardia, and convulsions [25,26]. These effects are likely due to intravascular injection, which also increases the theoretical risk of stroke from embolizing steroid particulates into the surrounding arteries, which include the SLA, STA, and ECA, among others [26]. Our findings suggest that variations of the relationship of the ibSLN to the SLA likely contribute to these complications and warrant the need for increased awareness among surgeons to decrease the vulnerability of the nerve. In a study by Kiray et al. [2], the course of the ibSLN is reported in relation to the superior cervical ganglion, but not to its respective artery, the SLA. Our study found that the ibSLN is more commonly related superoposterior to the SLA, and therefore would be helpful to identify and isolate the nerve to minimize iatrogenic injury. However, there was variation upon stratification by sex, which revealed that the ibSLN was more commonly related inferoposterior in male cadavers and superoposterior in female cadavers. This information, in addition to the study by Kiray et al. [2], will better assist surgeons in preservation of the nerve.

A limitation of our study includes a smaller sample size. Because these cadavers were studied by first-year medical students prior to this study, some nerves were not amenable to dissection and identification due to injury. Additionally, only one cadaver was of African-American descent, while the remaining were Caucasian.

## 5. Conclusions

To conclude, our study summarizes the importance of the anatomical variations of both the ibSLN and ebSLN. In general, the information gained from our study further emphasizes the need for routine identification and isolation of the SLN and its branches during procedures of the anterior neck, such as thyroidectomies, ACDF, and anesthetic nerve blocks. The variation in the course of the ebSLN and ibSLN with their respective arteries makes it clear that additional attention is required for preservation of the nerve and avoiding compression, stretching, ligation, and iatrogenic injury during in-office SLN blockade of the nerve.

## Figures and Tables

**Figure 1 diagnostics-11-01243-f001:**
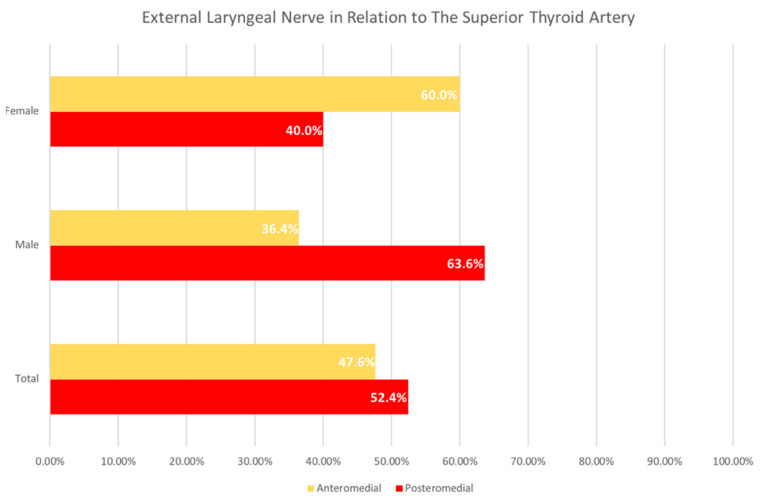
The Relationship of the External Laryngeal Nerve to the Superior Thyroid Artery. Analysis of the frequency based on the relationship of the external laryngeal nerve (ebSLN) to the superior thyroid artery (STA) showed, out of 21 samples, 52.4% of the ebSLN branches are related posteromedial to the STA, while 47.6% are related anteromedial to it. When stratified by sex, the ebSLN is more commonly related posteromedial to the STA in male cadavers (63.6%) compared to anteromedial relationship, which is more common in female cadavers (60.0%).

**Figure 2 diagnostics-11-01243-f002:**
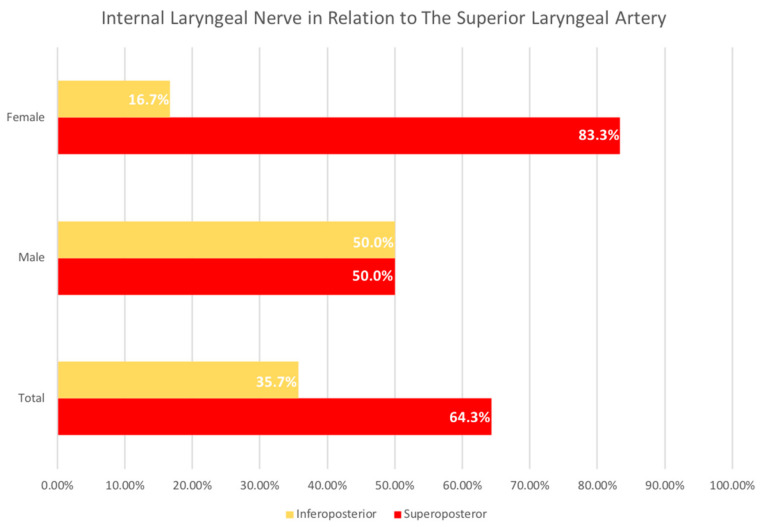
The Relationship of the Internal Laryngeal Nerve to the Superior Laryngeal Artery. Analysis of the frequency based on the relationship of the internal laryngeal nerve (ibSLN) to the superior laryngeal artery (SLA) showed, out of 14 samples, 64.3% of the ibSLN are related superoposterior to the SLA, while 35.7% are related inferoposterior to it. When stratified by sex, the ibSLN is more commonly related superoposterior to the SLA in female cadavers (83.3%) compared to inferoposterior relationship, which is more common in male cadavers (50.0%).

**Figure 3 diagnostics-11-01243-f003:**
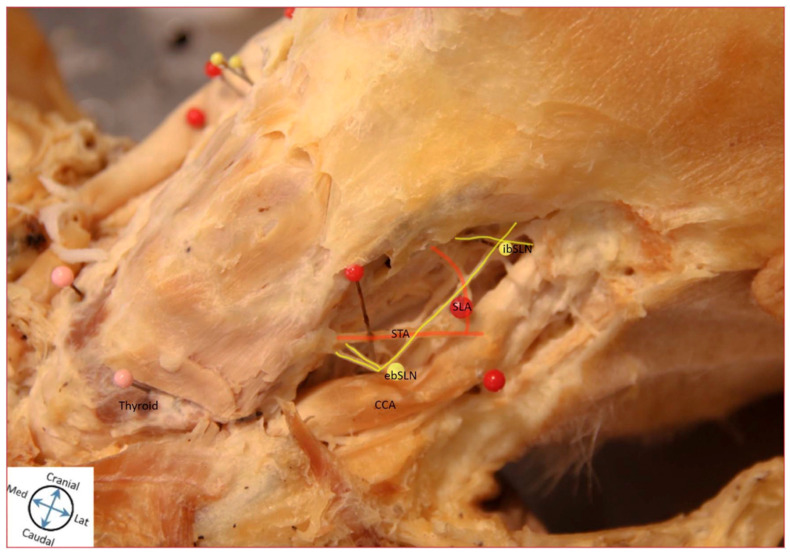
Cadaveric Representation of Anatomical Relationships. Cadaveric Image Representing Anatomical Relationship of External Laryngeal Nerve (ebSLN) and the Superior Thyroid Artery (STA) and Internal Laryngeal Nerve (ibSLN) with the Superior Laryngeal Artery (SLA). The image shows a unique course of the ebSLN in which it travels anterior to the STA and inserts posteriorly. The ibSLN is related superoposterior relative to the SLA. Abbreviations: ebSLN, external laryngeal nerve; ibSLN, internal laryngeal nerve; CCA, common carotid artery; ECA, external carotid artery; STA, superior thyroid artery; SLA, superior laryngeal artery.

**Figure 4 diagnostics-11-01243-f004:**
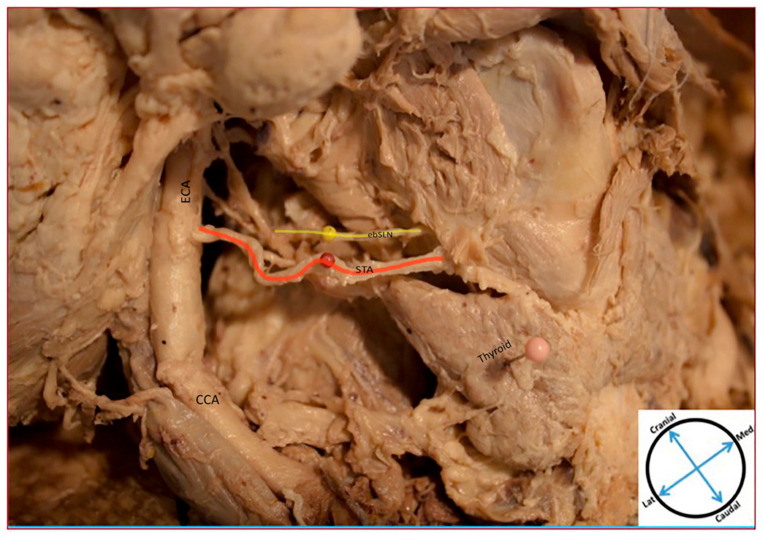
Cadaveric Representation of Anatomical Relationships. Cadaveric Image Representing Anatomical Relationship of External Laryngeal nerve (ebSLN) and the Superior Thyroid Artery (STA). The image shows the ebSLN traveling posteromedial to the STA; ibSLN not depicted. Abbreviations: ebSLN, external laryngeal nerve; CCA, common carotid artery; ECA, external carotid artery; STA, superior thyroid artery.

**Figure 5 diagnostics-11-01243-f005:**
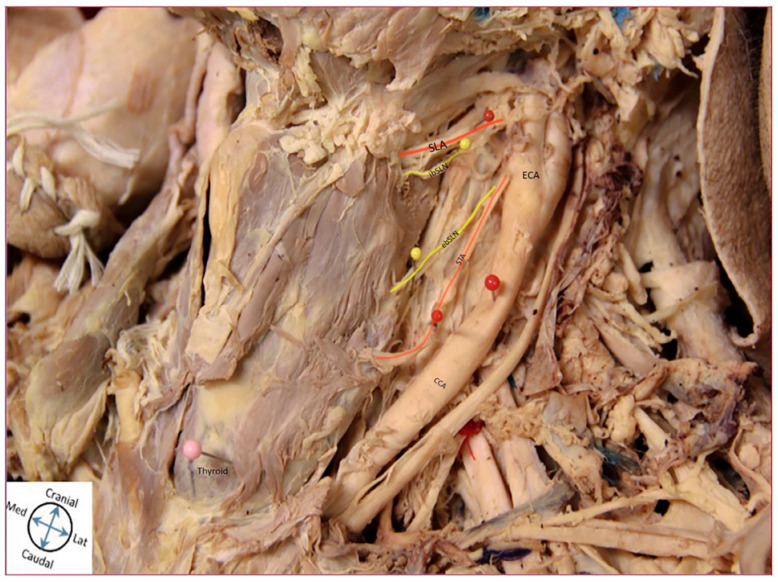
Cadaveric Representation of Anatomical Relationships. Cadaveric Image Representing Anatomical Relationship of External Laryngeal Nerve (ebSLN) and the Superior Thyroid Artery (STA) and Internal Laryngeal Nerve (ibSLN) with the Superior Laryngeal Artery (SLA). The image shows the ebSLN traveling anteromedial relative to the STA. The ibSLN travels inferoposterior relative to the SLA, which is directly branching off the ECA. Abbreviations: ebSLN, external laryngeal nerve; ibSLN, internal laryngeal nerve; CCA, common carotid artery; ECA, external carotid artery; STA, superior thyroid artery; SLA, superior laryngeal artery.

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
