# Peer review of "The Superior Laryngeal Nerve and Its Vulnerability in Surgeries of the Neck"

_diagnostics, 2021, doi:10.3390/diagnostics11071243_

Round 1
Reviewer 1 Report
This study by Antonio Dekhou and coauthors inspected 15 male and 20 female cadavers for variations of the external branch of the superior laryngeal nerve relative to the superior thyroid artery, and the internal branch of the superior laryngeal nerve relative to the superior laryngeal artery. They found a great deal of variability in nerve-to-artery relationships; almost equal frequency of posteromedial and anteromedial relationships between the external laryngeal nerve and the superior thyroid artery was found. Conversely, the internal laryngeal nerve was found in the superoposterior relationship to the superior laryngeal artery almost twice as frequently compared to the inferoposterior orientation. The data is presented in two basic plots and three good quality photographs. The manuscript is very well written, including a salient example of a famous patient who suffered loss of upper register of her singing voice after injury to the external branch of the superior laryngeal nerve; underscoring the importance of identification and isolation of these laryngeal nerve branches to prevent nerve injury.
The authors had a modest, but sufficient sample size in both male and female cadavers to analyze nerve-to-artery relationships in men and women separately. This information would strengthen the present study substantially, and either outcome - significant sex differences or not - will be of interest to the readership of the Journal. Publication in Diagnostics is recommended after addressing this deficiency and presenting sex-specific data in Figure 1 and 2.
Author Response
Dear Reviewer, thank you for the valuable and constructive comments. Based on your comments we have made the following changes:
Figure 1 and 2 have been changed to reflect the frequency of variation in the total sample size, in female and male samples. Additionally, we have reflected these changes in the first paragraph of the results section which now reads: “Of the 35 total cadavers, 20 were male and 15 were female. Twenty cadavers were Caucasian and one was African-American. Out of 21 cadavers in whom the ebSLN was dissected, 52.4% of the ebSLN are related posteromedial to the STA, while 47.6% are related anteromedial to it (Figure 1). Stratified by sex, the ebSLN is more commonly related posteromedial to the STA in male cadavers (63.6%) compared to female cadavers (40.0%) (Figure 1). Conversely, the ebSLN is more commonly related anteromedial to the STA in female cadavers (60.0%) compared to male cadavers (36.4%) (Figure 1). Out of 14 cadavers in whom the ibSLN was dissected, 64.3% of the ibSLN are related superoposterior to the SLA, while 35.7% are inferoposterior to it (Figure 2). Stratified by sex, the ibSLN is more commonly related superoposterior to the SLA in female cadavers (83.3%) compared to male cadavers (50.0%) (Figure 2). Conversely, the ibSLN is more commonly related inferoposterior to the SLA in male cadavers (50.0%) compared to female cadavers (16.7%) (Figure 2).”
The legends for Figure 1 and 2 have also been edited to reflect these changes and now read: “Figure 1. The Relationship of the External Laryngeal Nerve to the Superior Thyroid Artery. Analysis of the frequency based on the relationship of the external laryngeal nerve (ebSLN) to the superior thyroid artery (STA) showed, out of 21 samples, 52.4% of the ebSLN branches are related posteromedial to the STA, while 47.6% are related anteromedial to it. When stratified by sex, the ebSLN is more commonly related posteromedial to the STA in male cadavers (63.6%) compared to anteromedial relationship more common in female cadavers (60%).”
“Figure 2. The Relationship of the Internal Laryngeal Nerve to the Superior Laryngeal Artery. Analysis of the frequency based on the relationship of the internal laryngeal nerve (ibSLN) to the superior laryngeal artery (SLA) showed, out of 14 samples, 64.3% of the ibSLN are related superoposterior to the SLA, while 35.7% are related inferoposterior to it. When stratified by sex, the ibSLN is more commonly related superoposterior to the SLA in female cadavers (83.3%) compared to inferoposterior relationship common in male cadavers (50%).”
The discussion has been reflected these changes with the addition of the following:
“…. However, it is crucial to emphasize the variation present upon stratification by sex, which displayed that the ebSLN was more commonly related posteromedial in males and anteromedial in females.”
“… However, there was variation upon stratification by sex, which revealed that the ibSLN was more commonly related inferoposterior in male cadavers and superoposterior in female cadavers…”
Reviewer 2 Report
The anatomical relationships of the superior laryngeal nerve are analyzed in 36 cadavers.
The results may be useful for further investigation and for head and neck surgeons, especially during thyroidectomy.
The work is globally correct and has practical utility.
Author Response
Dear Reviewer, thank you for your valuable and constructive comments on our manuscript! We really appreciated it!